# Commercial System Reform, Enterprise Green Innovation and Enterprise ESG Performance

**Hui Li** [1] , **Dongsheng Yu** [2,3,*] **and Zhixuan Ke** [1]

1 School of Law and Business, Wuhan Institute of Technology, No. 206 Guanggu 1st Road, Wuhan 430205, China; zuellh2021@163.com (H.L.); wukzxlist@hotmail.com (Z.K.)
2 School of Statistics and Mathematics, Zhongnan University of Economics and Law, No. 182 Nanhu Avenue, Wuhan 430073, China
3 Postdoctoral Station of Statistics, Zhongnan University of Economics and Law, No. 182 Nanhu Avenue, Wuhan 430073, China
* Correspondence: zuelyds@126.com

**Abstract:** Taking the commercial system reform implemented in recent years as a "quasi-natural experiment", this paper systematically examines the relationship between commercial system reform and enterprise ESG performance and analyzes the intermediary effect of enterprise green innovation between the two, based on the data of China's commercial system reform and A-share listed companies in 2011–2021. The results show that the implementation of commercial system reform improves the level of ESG performance of enterprises, and green innovation of enterprises plays a positive role in the impact of commercial system reform on ESG performance of enterprises. After passing several robustness tests, the results of this paper are still valid. The results of the heterogeneity test show that the implementation of commercial system reform plays a greater role in promoting the ESG performance of state-owned enterprises, high-tech enterprises, high-financing-constraint enterprises, and high-pollution enterprises. The conclusions of this paper provide certain enlightenment for further promoting the reform of the commercial system and the construction of the ESG system and promoting enterprises to improve the level of modern green governance.

**Keywords:** commercial system reform; enterprise ESG; enterprise green innovation; heterogeneity

## 1. Introduction

ESG (environment, social and governance), which stands for environment, society, and governance, is one of the measurement criteria for the sustainable development of listed companies [1]. Enterprise green innovation has the dual advantages of environmental protection and the development of a green economy, which helps to promote the green transformation of production and life and becomes a key element to ensure the coordinated and unified development of economic benefits and the ecological environment. Compared with traditional investment, enterprise green innovation has some characteristics, such as a long profit cycle, large capital demand and high uncertainty of income. Therefore, enterprise green innovation often faces higher investment risk.

In fact, investors' willingness to supply capital is often affected by issues such as information asymmetry and financing constraints, which result in inefficient capital allocation and thus limit enterprises' green innovation. Generally speaking, the green experience of corporate CEOs can positively promote corporate green innovation [2]. From the external perspective of enterprises, stable and precise government economic and environmental policies are conducive to corporate green innovation [3], and the development of green finance is also conducive to corporate green innovation [4], thus ensuring the continuous progress of green, low-carbon and sustainable development of society. Enterprises can gain new market competitive advantages by valuing ESG practices and building ESG

advantages [5]. Therefore, how to improve the ability to fulfill ESG responsibilities has become an unavoidable topic for the high-quality development of enterprises.

The analysis of factors affecting the ESG performance of enterprises can be carried out from the macro and micro perspectives. From a macro perspective, the factors that affect the ESG performance of enterprises can be further subdivided into the following four points: policy system, economic system, cultural system and legal system. From the perspective of policies and systems, audit quality plays a positive role in improving the quality of ESG information disclosure [6]. From the perspective of the economic system, countries with a higher degree of economic development, a market-oriented financial system and a high market index of social responsibility usually have better ESG performance [7]. In terms of culture, cultural factors such as power distance, religion and harmony, equality and autonomy also affect the effect of enterprise ESG practice [8,9]. From the perspective of the legal system, the ESG level of enterprises in case law countries is often lower than that in civil law countries [10].

From a micro perspective, the factors affecting ESG performance can be roughly divided into company characteristics and executive characteristics. In terms of company characteristics, those companies with cross-listing, larger scale, more free cash flow, and more advertising investment usually show a higher level of ESG practice effect [11,12]. From the perspective of executive characteristics, young CEOs, female CEOs, CEOs who donate to political parties and CEOs who frequently appear in the media tend to devote more energy to ESG practice, and their companies will also have better ESG practice performance [13]. In addition, some scholars have found that the political orientation of executives will also affect the effect of ESG practice in enterprises. When the executives are Democrats, they tend to invest more in the ESG of the enterprise, thus improving the ESG practice performance of the enterprise [14].

The reform of the commercial system is one of the important means of domestic market economy reform. The existing literature mainly focuses on the macroeconomic effect of commercial system reform, such as the impact of commercial system reform on market entry rate, company system cost, foreign investment attraction, urban innovation level, etc. Some studies have pointed out that commercial system reform is conducive to reducing the institutional cost of companies [15] and is the main factor in improving the market entry rate [16]. Xia and Liu found that commercial system reform can improve the innovation level of cities [17]. Recently, scholars have begun to explore the micro-impact of commercial system reform on enterprises. Li and Yu found that commercial system reform can significantly reduce the negative impact of the two intermediary factors, namely institutional cost and entry cost, so as to increase the R&D funds and time of enterprises, and finally improve the innovation performance of enterprises [18]. Li et al. found that the promotion effect of commercial system reform on the total factor productivity of enterprises mainly comes from the reduction of institutional transaction costs [19]. However, up to now, there is no relevant research in the literature on the impact of commercial system reform on the ESG performance of micro-enterprises.

In view of this, this paper takes the commercial system reform implemented in recent years as a "quasi-natural experiment", matches the hand-collected data of commercial system reform with the corporate data of A-share listed companies from 2011 to 2021, explores the causal relationship between commercial system reform and corporate ESG performance. Compared with the existing literature, the marginal contributions of this paper are as follows.

First, unlike the previous literature on the impact of administrative approval reform on enterprises, this paper studies how commercial system reform affects the ESG performance of micro-listed enterprises and the mediating role of corporate green innovation in it. Secondly, this paper adopts the methods of experimental group and control group, takes the implementation of commercial system reform in a certain city at a certain time as the experimental sample, and uses the DID empirical method to evaluate the impact of commercial system reform on the ESG performance of listed enterprises. Finally, this paper

deeply analyzes the mechanism of corporate green innovation in the impact of commercial system reform on corporate ESG performance and also discusses the heterogeneity of commercial system reform measures and policy effects among different firms. These studies are of great significance for promoting the reform of commercial systems and the construction of ESG systems.

The remaining parts of this paper are arranged as follows: the second part is the Commercial System Reform and Research Hypothesis; the third part is the Research Design; the fourth part is Empirical Analysis; the fifth part is the Conclusions and Policy Implications; and the last part is the Limitations.

## 2. Commercial System Reform Background and Research Hypothesis

(1)    Background of commercial system reform

The commercial system is the policy for market economic activities such as the emergence, existence, and withdrawal of market entities, and the guarantee for the establishment, production, and operation of enterprises. However, over the past four decades of reform and opening up, the commercial system has lagged behind the reform of China's socialist market economic system. The complicated conditions for commercial registration have replaced post-supervision with administrative approval for registration and establishment, which has restricted the free flow of production factors, reduced the efficiency of market resource allocation, and hindered high-quality economic development.

The Several Opinions on Promoting Fair Market Competition and Maintaining Normal Market Order, issued by The State Council in 2014, launched the reform of the commercial system. In 2015, the reform of the commercial system continued to advance, and relevant state departments issued the Opinion on accelerating the reform of the "three certificates in one" registration system, and the Opinion on strengthening the supervision during and after the reform of "license before license", actively promoting the registration process of "three certificates in one" and the registration method of "one according to one code", clarifying the principles of market supervision. Enterprise registration is more convenient, and an efficient, clear, and transparent regulatory system has been established during and after the event, correctly handling the relationship between the government and the market, and maintaining the market order of fair competition.

In 2016, The State Council issued the Notice on Accelerating the Reform of the Registration system of "Five Certificates in One, One License in One Code". On the basis of effectively implementing the reform of the registration system of three certificates in one, the two registration certificates of statistics and social insurance were added to achieve "one license in one code" and "five certificates in one code", guiding and supporting mass innovation, updating and improving the business environment. Then, various provinces and cities have carried out a series of reform policies such as enterprise registration, supervision, and cancellation, such as the whole electronic reform, one Netcom office, "Internet + supervision", one window acceptance, "double random, one open" supervision, one card and so on. Over the past six years, the commercial system has been continuously contributing to China's development, correctly handling the relationship between the government and the market, and actively promoting the formation of an "efficient market + successful government" system.

(2)    Research hypothesis

From the perspective of corporate environmental responsibility performance, it is necessary for enterprises to increase their investment in green research and development, improve the production process and enhance the level of green innovation [20]. From the perspective of corporate social responsibility performance, a good business environment is conducive to the formation of close links between enterprises and various stakeholders, alleviating the problem of information asymmetry, thereby increasing investor confidence and obtaining resources required for corporate ESG practice, and ultimately improving corporate performance [21]. The reform of the commercial system is the inheritance and

development of the reform of administrative examination and approval. It has played an important role in part of the system and order of China's socialist market economy.

From the perspective of the reputation hypothesis, when market competition is relatively fierce, in order to avoid or transfer market risks, enterprises will actively perform ESG responsibilities to build a good corporate image so as to alleviate the risks and pressures brought by market competition [22]. According to the signal transmission theory, an enterprise's active performance of ESG responsibilities can not only establish a good corporate image, but also convey positive signals of good business status to the outside world, thus enhancing the trust of stakeholders [23]. Therefore, in order to alleviate the external regulatory pressure from the securities market, enterprises in highly competitive industries tend to obtain continuous competitive advantages by improving their ESG performance. Commercial system reform can guide and serve market subjects to take the initiative to correct and repair illegal and dishonest behaviors so as to avoid financing restrictions, so that enterprises can have sufficient funds to carry out ESG responsibility practice activities [24].

The core pursuit of corporate green innovation is to achieve green development. In order to fully leverage the innovation capabilities of enterprises, it is necessary to reset resources. Enterprises can achieve low-carbon and high-tech transformation of their industries by adjusting their industrial structure, providing a continuous source of power for subsequent environmental improvement [25]. In addition, the strengthening of green innovation technology also reduces energy intensity, reduces energy consumption in production links, has a positive impact on the environment and improves environmental quality by reducing extensive dependence on energy [26,27].

From the perspective of legitimacy theory and stakeholder theory, there is an implicit social contract between enterprises and stakeholders. If enterprises deviate from this contract, they will face the legitimacy question, litigation risk and pressure from public opinion [28]. Institutional investors play an important role among stakeholders. They provide financial resources for enterprises, regulate corporate behavior, require enterprises to actively comply with social expectations and legal requirements [29], and supervise the performance of enterprises in environmental, social responsibility and corporate governance. Therefore, constrained by the pressure of the capital market, enterprises have the motivation to innovate in green technology. Enterprises will actively implement environmental management decisions to comply with universally recognized values and ethical norms among investors, meet consumers' requirements for new-era value orientation, and maintain their corporate image and reputation. [30]. The reform of the commercial system can help to save the cost of enterprises, increase the investment of enterprises in research and development, reduce pollution emissions, and maintain the image and reputation of enterprises, thus improving the ESG of enterprises.

Based on the above analysis, H1 is proposed: the reform of the commercial system promotes the ESG performance of enterprises.

Based on the above analysis, H2 is proposed: enterprise green innovation plays a positive role in the impact of commercial system reform on enterprise ESG performance.

## 3. Research Design

(1) Model building

Based on the methods of Zhang et al. [31], this paper adopts the multi-stage differentially differential model for empirical evaluation. The specific model construction is as follows:

$$ESG_{etc} = a_0 + a_1 Policy_{ect} + \beta X_{ect} + \eta_e + \delta_c + \mu_t + \epsilon_{ect} \tag{1}$$

In Equation (1), the subscript $e$ is an A-share listed enterprise, $c$ is a city variable, $t$ is a year variable, $ESG_{etc}$ is an ESG performance variable of A-share listed enterprises, and $Policy_{ect}$ is a virtual variable of commercial system reform. If a city begins to implement commercial system reform in a certain year, $Policy_{ect}$ is assigned to 1; otherwise, $Policy_{ect}$ is assigned to 0. $X_{ect}$ is a control variable at the city and listed enterprise levels, $\eta_e$ is the fixed

effect of the enterprise, $\delta_c$ is the urban fixed effect, $\mu_t$ is the fixed effect of the year, $\epsilon_{ect}$ is the disturbance term.

(2)    Variable selection

(a) Explained variable

Based on the methods of Yuan et al. [32], this paper adopts the Huazheng ESG index to evaluate enterprises' ESG performance quantitatively in benchmark regression, intermediary mechanism test, and heterogeneity analysis and adopts the Bloomberg ESG index to conduct a robustness test.

(b) Core explanatory variable

Based on the methods of Li et al. [19], this paper takes pilot cities of commercial system reform as reform samples. The pilot cities of commercial system reform are taken as the experimental group, while the other cities are taken as the control group. In other words, the variable *Policy* in the above equation is set to a city selected as a pilot city in a certain year, then the city will be assigned a value of 1 in that year and subsequent years, and the rest will be assigned a value of 0.

(c) Control variable

This paper uses for reference Enterprise Size (Size), the method from Zhang et al. [33], and evaluates the total assets at the end of the year by taking the logarithm. Asset-liability ratio (Lev), based on the method adopted by Li et al. [34], is measured by the ratio of the total liabilities of the company to the total assets at the end of the year. Cashflow, the method of Ai et al. [35], is used for reference in this paper to measure the ratio of net cash flow generated by operating activities to total assets. Profitability (ROA), based on the method adopted by Rahman et al. [36], is evaluated by the ratio of corporate net profit to total average assets. The size of directors (Board), based on the method of Phuong and Hung [37], is evaluated by taking the natural logarithm of the total number of directors. The ratio of independent directors (Indep), the method from Mediaty [38], is measured by the ratio of the number of independent directors to the total number of directors. Listing years (ListAge), the method of Shi and Shen [39], this paper adds the listing years of the company by 1 and then takes the natural logarithm to evaluate. Foreign direct investment (fdi), this paper draws on the method of Li et al. [40] to evaluate the listage by using the logarithm of FDI. Economic development (pgdp), based on the method adopted by Li and Yu [18], this paper takes the logarithm of the per capita GDP of each city for evaluation. The detailed definitions of variables in this paper are shown in Table 1.

(3)    Data description

Processing of all data in this paper: 1) Manually collect the data of commercial system reform in all prefecture level and above cities in China from 2011 to 2021 on the official websites of municipal administrations for Industry and Commerce and the government, and sort out and match the data of commercial system reform in the cities where listed enterprises are located (it mainly includes "one license and one code" and "multi certificate integration", as well as the announcement time of "registered capital registration system reform"). 2) Using the method of Yu and Luu [12] as a reference, we processed the data of listed enterprises from 2011 to 2021, deleted the samples of incomplete core variables and financial companies, and excluded the samples of financial outliers (including total assets less than 0, net assets less than 0, asset-liability ratio greater than 1 and abnormal operation samples). 3) We match the relevant data of commercial system reform in cities at the prefecture level and above with the data of A-share listed enterprises according to the address of listed enterprises to obtain 9207 observed values. The descriptive statistics of the variables in this paper are shown in Table 2.

**Table 1.** Definition of variables.

| Name | Type | Description |
|---|---|---|
| ESG | Explained variable | The Huazheng ESG index is used in benchmark regression, while the Bloomberg ESG index is used in robustness regression |
| Reform of the commercial system (Policy) | Explanatory variable | The city that implemented the commercial system reform in a certain year was assigned a value of 1, while the rest were assigned a value of 0 |
| Size of the enterprise (Size) | Control variable | Log of total assets at the end of the year |
| Asset liability ratio (Lev) | Control variable | The ratio of the company's total liabilities to total assets at the end of the period |
| Enterprise's own turnover capacity (Cashflow) | Control variable | The ratio of net cash flow formed by operating activities to total assets |
| Profitability (ROA) | Control variable | The ratio of net profit to average total assets |
| Size of directors (Board) | Control variable | The natural logarithm of the total number of directors on the board of directors |
| Proportion of independent directors (Indep) | Control variable | The ratio of the number of independent directors to the total number of directors on the board |
| Listing life (ListAge) | Control variable | The logarithm of the company's listing years plus 1 |
| Foreign direct investment (fdi) | Control variable | The actual use of foreign direct investment |
| Economic development (pgdp) | Control variable | Log of per capita GDP of each city |

**Table 2.** Descriptive statistics of main variables.

| Variables | (1)<br>N | (2)<br>Mean | (3)<br>SD | (4)<br>Min | (5)<br>Max |
|---|---|---|---|---|---|
| ESG | 9207 | 4.439 | 1.034 | 1 | 7.750 |
| Policy | 9207 | 0.208 | 0.406 | 0 | 1 |
| Size | 9207 | 23.22 | 1.279 | 19.55 | 26.43 |
| Lev | 9207 | 0.486 | 0.198 | 0.0310 | 0.925 |
| ROA | 9207 | 0.0482 | 0.0640 | −0.398 | 0.254 |
| Cashflow | 9207 | 0.0585 | 0.0693 | −0.200 | 0.257 |
| Board | 9207 | 2.175 | 0.201 | 1.609 | 2.708 |
| Indep | 9207 | 0.376 | 0.0555 | 0.286 | 0.600 |
| ListAge | 9207 | 2.507 | 0.663 | 0 | 3.367 |
| PGDP | 9207 | 11.18 | 0.480 | 9.706 | 12.12 |
| fdi | 9207 | 12.44 | 1.382 | 6.588 | 15.33 |

Note: the data was analyzed using Stata16 software.

## 4. Empirical Analysis

(1)  Benchmark regression results

Table 3 reports the benchmark regression results of the impact of business system reform on enterprise ESG, with the dependent variable being enterprise ESG. Equations (1)–(3) in Table 3 are estimated results without considering control variables. Equations (4)–(6) in Table 3 represent the estimated results when considering control variables, where Equations (1) and (4) do not control for individual and time effects, Equations (2) and

(5) control for fixed effects of city and time, while Equations (3) and (6) control for fixed effects of year, city and enterprise. It is not difficult to find that whether control variables are added or different fixed effects are controlled, the estimation coefficient of commercial system reform (Policy) is significantly positive, which shows that the implementation of commercial system reform helps enterprises to carry out ESG practice activities, improves the ESG rating index of enterprises, so H1 is supported. On the one hand, the reform of the commercial system can reduce the time for enterprises to start up and approve, reduce the expenditure on non-productive activities, and allocate more expenditure to production factors and scientific and technological research and development [41], which is conducive to the improvement of the motivation and level of ESG implementation. On the other hand, it also reduces the threshold for enterprises to enter the market, increases the probability of enterprise entry, and puts incumbent enterprises under greater entry threats and competitive pressure, forcing incumbent enterprises to update production technology, actively carry out research and innovation activities [18], eliminate enterprises with low ESG, and ultimately promote the realization of enterprise ESG responsibility.

**Table 3.** Benchmark region results.

| Variables | (1) ESG | (2) ESG | (3) ESG | (4) ESG | (5) ESG | (6) ESG |
|---|---|---|---|---|---|---|
| Policy | 0.0694 *** | 0.0560 ** | 0.0296 ** | 0.2926 *** | 0.2146 *** | 0.1871 *** |
| | (0.024) | (0.023) | (0.012) | (0.027) | (0.034) | (0.027) |
| Size | | | | 0.2165 *** | 0.2434 *** | 0.2552 *** |
| | | | | (0.010) | (0.011) | (0.025) |
| Lev | | | | −0.6111 *** | −0.6440 *** | −0.9481 *** |
| | | | | (0.072) | (0.075) | (0.103) |
| Cashflow | | | | −0.8102 *** | −0.5919 *** | −0.5995 *** |
| | | | | (0.170) | (0.166) | (0.152) |
| ROA | | | | 1.7908 *** | 1.3725 *** | 0.6581 *** |
| | | | | (0.230) | (0.223) | (0.214) |
| Board | | | | 0.1513 ** | 0.1707 *** | 0.1387 |
| | | | | (0.061) | (0.063) | (0.091) |
| Indep | | | | 1.6824 *** | 1.5836 *** | 1.8613 *** |
| | | | | (0.210) | (0.221) | (0.266) |
| ListAge | | | | −0.0150 | 0.0162 | 0.1144** |
| | | | | (0.017) | (0.018) | (0.048) |
| Constant | 4.4243 *** | 4.4268 *** | 4.4447 *** | −2.8070 *** | −2.6276 *** | −3.4645 *** |
| | (0.013) | (0.011) | (0.008) | (0.354) | (0.781) | (0.778) |
| City control variables | no | no | no | yes | yes | yes |
| Time fixed effect | no | yes | yes | no | yes | yes |
| City fixed effect | no | yes | yes | no | yes | yes |
| Firm fixed effect | no | no | yes | no | no | yes |
| R-squared | 0.051 | 0.185 | 0.642 | 0.099 | 0.254 | 0.660 |
| Observations | 9207 | 9200 | 9133 | 9207 | 9200 | 9133 |

Note: *** and ** mean significant at the level of 1% and 5%, respectively, and the values in brackets represent the robust standard error of urban level clustering.

(2)   Robustness test

(a) Parallel trend policy timing changes

In order to ensure the reliability of the research conclusions, this paper advances the policy impact time of commercial system reform by one year, two years, one year, and two years, respectively. Table 4 reports the regression results of the parallel trend test. Equations (1)–(3) in Table 3 are the estimation results without considering control variables, and Equations (4)–(6) are the estimation results considering control variables, among them, Equations (1) and (4) do not control individual and time effects, Equations (2) and (5) control city and enterprise fixed effects, and Equations (3) and (6) control year, city and enterprise fixed effects. It is not difficult to find whether control variables are added or different

fixed effects are controlled, in the first two years of the implementation of the commercial system reform, the estimated coefficient of Policy$^{-2}$ and Policy$^{-1}$ is not significant at the conventional significance level; that is, the parallel trend hypothesis is valid. In the two years after the implementation of the commercial system reform, the estimated coefficient of Policy$^1$ and Policy$^2$ is significant at the 1% significance level, which indicates that the significant improvement of ESG after the implementation of the commercial system reform is not caused by the prior differences of enterprises. The conclusion that the commercial system reform obtained in the benchmark regression model helps to improve the ESG of enterprises is credible.

**Table 4.** Parallel trend policy timing change.

| Variables | (1) ESG | (2) ESG | (3) ESG | (4) ESG | (5) ESG | (6) ESG |
|---|---|---|---|---|---|---|
| Policy$^{-2}$ | 0.0058 | −0.0634 | 0.0167 | −0.0045 | −0.0688 | 0.0764 |
| | (0.026) | (0.098) | (0.077) | (0.025) | (0.101) | (0.077) |
| Policy$^{-1}$ | 0.0040 | −0.1112 | 0.0236 | −0.0125 | −0.1118 | 0.1026 |
| | (0.026) | (0.097) | (0.060) | (0.026) | (0.069) | (0.092) |
| Policy$^1$ | 0.0042 ** | 0.1260 ** | 0.0051 *** | 0.0345 *** | 0.1514 *** | 0.0898 * |
| | (0.002) | (0.055) | (0.001) | (0.008) | (0.058) | (0.050) |
| Policy$^1$ | 0.0147 ** | 0.1071 ** | 0.0021 ** | 0.0376 *** | 0.1415 ** | 0.0930 ** |
| | (0.007) | (0.053) | (0.001) | (0.009) | (0.056) | (0.047) |
| Control variable | no | no | no | yes | yes | yes |
| Time fixed effect | no | yes | yes | no | yes | yes |
| City fixed effect | no | yes | yes | no | yes | yes |
| Firm fixed effect | no | no | yes | no | no | yes |
| Observations | 9207 | 9200 | 9133 | 9207 | 9200 | 9133 |

Note: ***, ** and * mean significant at the level of 1%, 5% and 10%, respectively, and the values in brackets represent the robustness standard error of urban-level clustering.

(b) Replace the explained variable

We further adopted the Bloomberg ESG index for the robustness test, and the regression results are shown in Table 5. It is not difficult to find that no matter whether control variables are added or different fixed effects are controlled, the estimated coefficients of Policy are significantly positive, which further confirms that the implementation of commercial system reform is conducive to enterprises' ESG practice activities and improves enterprises' ESG rating index, supporting the research conclusions obtained in the benchmark regression of this paper.

(c) A placebo test

Referring to the research of Chetty et al. [42], this paper adopts an indirect placebo test: we randomly generate a set list of market authorities to produce an incorrect estimate of the multiplier coefficient value and repeat the process 500 times to produce a corresponding 500 coefficient estimate. Obviously, if non-observational factors do not have a significant impact on the "quasi-natural experiment" of real commercial system reform, then the multiplier coefficient of the impact of the above randomly generated "quasi-natural experiment" on enterprise ESG performance activities should meet the average value of 0. The distribution of the estimated coefficient shown in Figure 1 shows that the average of the coefficient estimates is around 0, indicating that there is no serious problem of missing variables in the model setting, which can be considered to have passed the placebo test, and the core conclusion is still robust.

**Table 5.** Substituting explained variables.

| Variables | (1) ESG | (2) ESG | (3) ESG | (4) ESG | (5) ESG | (6) ESG |
|---|---|---|---|---|---|---|
| Policy | 0.3579 *** (0.006) | 0.3491 *** (0.006) | 0.3573 *** (0.005) | 0.2148 *** (0.006) | 0.0731 *** (0.008) | 0.0773 *** (0.006) |
| Size | | | | 0.0932 *** (0.002) | 0.0890 *** (0.002) | 0.0657 *** (0.006) |
| Lev | | | | −0.2238 *** (0.014) | −0.2104 *** (0.014) | −0.2220 *** (0.021) |
| Cashflow | | | | −0.3713 *** (0.034) | −0.2635 *** (0.033) | −0.0587 * (0.030) |
| ROA | | | | 0.1945 *** (0.041) | 0.1091 *** (0.040) | 0.0033 (0.038) |
| Board | | | | 0.0398 *** (0.012) | 0.0478 *** (0.012) | 0.0183 (0.019) |
| Indep | | | | 0.1430 *** (0.042) | 0.1184 *** (0.044) | 0.1518 *** (0.058) |
| ListAge | | | | 0.0191 *** (0.003) | 0.0051 (0.004) | 0.0588 *** (0.011) |
| Constant | 3.3782 *** (0.003) | 3.3763 *** (0.003) | 3.3782 *** (0.002) | −0.3345 *** (0.078) | −3.5936 *** (0.162) | −3.2514 *** (0.172) |
| City control variables | no | no | no | yes | yes | yes |
| Time fixed effect | no | yes | yes | no | yes | yes |
| City fixed effect | no | yes | yes | no | yes | yes |
| Firm fixed effect | no | no | yes | no | no | yes |
| R-squared | 0.274 | 0.372 | 0.659 | 0.506 | 0.603 | 0.796 |
| Observations | 9207 | 9200 | 9133 | 9207 | 9200 | 9133 |

Note: *** and * mean significant at the level of 1% and 10%, respectively, and the values in brackets represent the robustness standard error of urban level clustering.

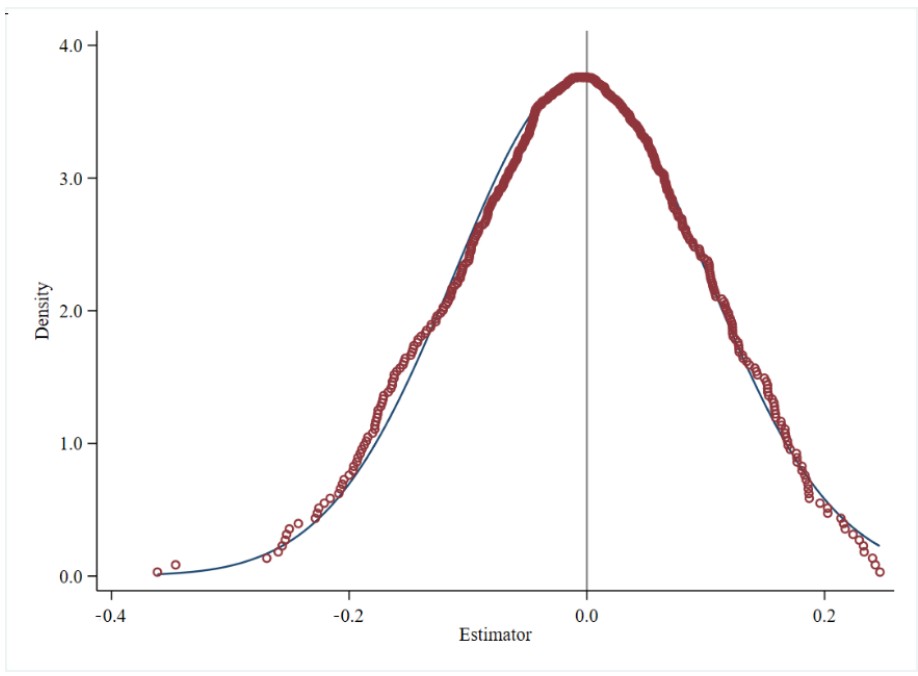

**Figure 1.** Placebo test results.

(d) PSM-DID

In this paper, PSM-DID is used to solve the possible randomness problem. In the matching link, we select enterprise size (Size), asset-liability ratio (Lev), enterprise's own turnover capacity (cashflow), profitability (ROA), director size, independent director ratio

(Indep), listing year (listage), foreign direct investment (FDI) and per capita GDP (PGDP) variable of each city is used as the matching variable, as shown in Figure 2. The regression results of did model after PSM treatment are shown in Table 6. The sample size at the national level has been reduced, but from the regression results, the reform of the commercial system will still significantly promote the ESG performance activities of enterprises. With the introduction of control variables, after controlling other factors, the direction and significance of the reform of the commercial system have not changed, both of which are positive and significant. The above results show that the analysis results of the previous multiphase model are robust.

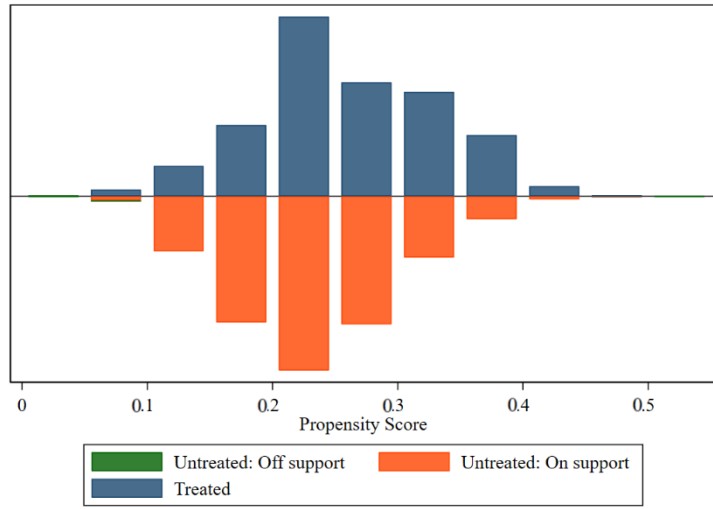

**Figure 2.** PSM-DID matching results.

(e) Other robustness tests

In order to further enhance the reliability of the research results, in addition to the above tests, this paper also carries out a series of other aspects of the robustness analysis of the benchmark regression results. The corresponding results are reported in Table 7. First of all, the sample period of this study covers 2011–2021, covering the period of the COVID-19 pandemic. In order to eliminate the influence of COVID-19 on the regression results, all data after 2020 were deleted for robustness analysis and reported in column (1) of Table 7. Secondly, in order to exclude the influence of extreme values in the samples, we conducted tail indentation at 1% subsites of the samples and reported them in column (2) of Table 7. Finally, in order to mitigate the influence of the endogeneity of control variables on the research results, the regression results of all control variables lagging one stage were further given and reported in column (3) of Table 7. It is not difficult to find that regardless of the setting method, the research conclusion that commercial system reform can help improve the ESG performance of enterprises is always valid.

(3)    Testing the intermediary mechanism of enterprise green innovation

As previously analyzed, enterprise green innovation plays a positive role in the impact of commercial system reform on enterprise ESG performance. In view of this, using the method of Yang and Yu [43] as a reference, this paper constructs an intermediary effect test model to identify the mechanism of enterprise green innovation in the process of commercial system reform affecting enterprise ESG performance. The test model is designed as follows:

$$ESG_{etc} = a_0 + a_1 Policy_{ect} + X_{ect} + \eta_1 + \delta_1 + \mu_1 + \epsilon_1$$

$$M_{etc} = \vartheta_0 + \vartheta_1 Policy_{ect} + X_{ect} + \eta_2 + \delta_2 + \mu_2 + \epsilon_2 \qquad (2)$$

$$ESG_{etc} = \tau_0 + \tau_1 Policy_{ect} + \tau_2 M_{etc} + X_{ect} + \eta_3 + \delta_3 + \mu_3 + \epsilon_3$$

Among them, the subscript $e$ is an A-share listed enterprise, $c$ represents the city variable, $t$ represents the year variable, $ESG_{etc}$ is an ESG performance variable of A-share listed enterprises, and $Policy_{ect}$ is a virtual variable of commercial system reform. If a city begins to implement commercial system reform in a certain year, $Policy_{ect}$ is assigned to 1; otherwise, $Policy_{ect}$ is assigned to 0. $M_{etc}$ is the intermediary mechanism variable and the green innovation of enterprises. This paper uses the number of green invention patent registrations of enterprises and R&D investment of enterprises to measure. $\eta_e$ is the fixed effect of the enterprise, $\delta_c$ is the urban fixed effect, $\mu_t$ is the fixed effect of the year, and $\epsilon_{ect}$ is the disturbance term.

**Table 6.** PSM-DID region results.

| Variables | (1) OLS ESG | (2) Fe ESG | (3) Weight Not Empty ESG | (4) On_Support ESG | (5) Weight_Reg ESG |
|---|---|---|---|---|---|
| Policy | 0.2926 *** (0.029) | 0.1871 *** (0.027) | 0.1332 *** (0.039) | 0.1872 *** (0.027) | 0.1342 *** (0.033) |
| Size | 0.2165 *** (0.010) | 0.2552 *** (0.022) | 0.2839 *** (0.032) | 0.2494 *** (0.022) | 0.2929 *** (0.027) |
| Lev | −0.6111 *** (0.071) | −0.9481 *** (0.096) | −1.0246 *** (0.137) | −0.9318 *** (0.096) | −0.9652 *** (0.117) |
| Cashflow | −0.8102 *** (0.171) | −0.5995 *** (0.143) | −0.4201 ** (0.201) | −0.5995 *** (0.144) | −0.3838 ** (0.173) |
| ROA | 1.7908 *** (0.214) | 0.6581 *** (0.187) | 0.5910 ** (0.268) | 0.6725 *** (0.187) | 0.4305 * (0.229) |
| Board | 0.1513 ** (0.060) | 0.1387 * (0.084) | 0.0943 (0.120) | 0.1377 (0.084) | 0.0320 (0.103) |
| Indep | 1.6824 *** (0.212) | 1.8613 *** (0.249) | 1.5265 *** (0.371) | 1.8542 *** (0.250) | 1.3386 *** (0.319) |
| ListAge | −0.0150 (0.017) | 0.1144 ** (0.045) | 0.0428 (0.066) | 0.1119 ** (0.046) | 0.0390 (0.055) |
| Constant | −2.8070 *** (0.371) | −3.4645 *** (0.721) | −2.4304 ** (1.074) | −3.3867 *** (0.725) | −2.0750 ** (0.917) |
| City control variables | yes | yes | yes | yes | yes |
| Time fixed effect | no | yes | yes | yes | yes |
| City fixed effect | no | yes | yes | yes | yes |
| Firm fixed effect | no | yes | yes | yes | yes |
| R-squared | 0.099 | 0.660 | 0.693 | 0.660 | 0.711 |
| Observations | 9207 | 9133 | 4871 | 9103 | 6354 |

Note: ***, ** and * mean significant at the level of 1%, 5% and 10%, respectively, and the values in brackets represent the robustness standard error of urban level clustering.

Table 8 reports the test results of the impact mechanism. In Table 8, column (1) is the benchmark regression result, and the estimated coefficients of commercial system reform in columns (2) and (4) are both positive, consistent with the conclusion drawn by the theoretical model that the reform of the commercial system has simplified the registration process, relaxed the conditions for industrial and commercial registration, and reduced the time for enterprise establishment and business processing [19]. It has promoted green patent applications of enterprises, expanded R&D investment in enterprise innovation [17] and improved the green innovation ability of enterprises [18]. The estimated coefficient of green patent applications and R&D investment of enterprises in columns (3) and (5) is significantly positive, indicating that green innovation in enterprises plays a positive promoting role in the impact of commercial system reform on ESG performance, so H2 is supported.

**Table 7.** Other robustness tests.

| Variables | (1) ESG 2011–2019 Data | (2) ESG Winsorize | (3) ESG Lagging One Stage |
|---|---|---|---|
| Policy | 0.2091 *** | 0.1876 *** | 0.1924 *** |
| | (0.029) | (0.027) | (0.028) |
| Size | 0.2213 *** | 0.2579 *** | 0.2584 *** |
| | (0.029) | (0.025) | (0.025) |
| Lev | −0.8784 *** | −0.9513 *** | −0.9494 *** |
| | (0.118) | (0.103) | (0.104) |
| Cashflow | −0.5892 *** | −0.6090 *** | −0.6093 *** |
| | (0.165) | (0.154) | (0.156) |
| ROA | 0.1605 | 0.6966 *** | 0.7163 *** |
| | (0.256) | (0.225) | (0.227) |
| Board | 0.1735 * | 0.1395 | 0.1369 |
| | (0.103) | (0.091) | (0.092) |
| Indep | 1.9605 *** | 1.8564 *** | 1.8971 *** |
| | (0.291) | (0.266) | (0.267) |
| ListAge | 0.1538 *** | 0.1185 ** | 0.1352 ** |
| | (0.059) | (0.052) | (0.055) |
| Constant | −3.6304 *** | −3.4866 *** | −3.5400 *** |
| | (0.978) | (0.783) | (0.786) |
| City control variables | yes | yes | yes |
| Time fixed effect | yes | yes | yes |
| City fixed effect | yes | yes | yes |
| Firm fixed effect | yes | yes | yes |
| R-squared | 0.669 | 0.660 | 0.657 |
| Observations | 7118 | 9133 | 8915 |

Note: ***, ** and * mean significant at the level of 1% and 5%, respectively, and the values in brackets represent the robustness standard error of urban level clustering.

(4)    Heterogeneity analysis

(a) Distinguishing the nature of property rights

The essence of commercial system reform is that the government should delegate more power, reduce intervention in the market and adjust the relationship with the market. Table 9 reports the results of the heterogeneous regression of commercial system reform on ESG sub-samples of state-owned enterprises and private enterprises. Equations (1) and (2) in Table 9 are the estimated results of state-owned enterprises, and the estimated coefficient of commercial system reform (Policy) is significantly positive, indicating that commercial system reform has promoted the ESG performance of state-owned enterprises. Equations (3) and (4) are the estimated results of private enterprises, and the estimated coefficient of commercial system reform (Policy) is also significantly positive. It is worth noting that the reform of the commercial system has greatly promoted the ESG performance of state-owned enterprises. On the one hand, state-owned enterprises have a close relationship with the government, have a certain government background and resource advantages, can receive government attention and support to a certain extent, and can more easily obtain preferential funds and resources provided by the government. These advantages can make it easier for state-owned enterprises to participate in environmental and social responsibility investment than private enterprises. On the other hand, the governance structure of state-owned enterprises is relatively perfect, and the management system is relatively standardized and rigorous. With the deepening of the reform of the commercial system, the transparency of state-owned enterprises has gradually improved, and information disclosure has become more and more extensive, which makes the ESG practice of state-owned enterprises more recognized and supported by the outside world than that of private enterprises.

**Table 8.** Impact mechanism test of green innovation capability of enterprises.

| Variables | (1) ESG | (2) Green Patent | (3) ESG | (4) R&D Investment | (5) ESG |
|---|---|---|---|---|---|
| Policy | 0.1871 *** (0.027) | 0.1084 ** (0.051) | 0.1819 *** (0.044) | 0.1162 ** (0.056) | 0.185 5*** (0.027) |
| Green patent | | | 0.0344 ** (0.015) | | |
| R&D investment | | | | | 0.0143 *** (0.005) |
| Size | 0.2552 *** (0.025) | 0.0786 * (0.047) | 0.2703 *** (0.039) | 0.1117 * (0.067) | 0.2536 *** (0.025) |
| Lev | −0.9481 *** (0.103) | 0.1161 (0.217) | −1.2057 *** (0.169) | −3.0793 *** (0.291) | −0.9042 *** (0.104) |
| Cashflow | −0.5995 *** (0.152) | −0.0809 (0.293) | −0.7216 *** (0.242) | -0.1336 (0.316) | −0.5976 *** (0.152) |
| ROA | 0.6581 *** (0.214) | 0.5567 (0.365) | −0.1014 (0.318) | −7.1161 *** (0.621) | 0.7596 *** (0.221) |
| Board | 0.1387 (0.091) | −0.0702 (0.163) | 0.2153 (0.141) | −0.5253** (0.225) | 0.1462 (0.091) |
| Indep | 1.8613 *** (0.266) | −0.7179 (0.479) | 2.0935 *** (0.435) | −1.5415 *** (0.592) | 1.8833 *** (0.266) |
| ListAge | 0.1144 ** (0.048) | 0.2269 ** (0.114) | 0.1875 ** (0.086) | −0.0434 (0.116) | 0.1151 ** (0.048) |
| Constant | −3.4645 *** (0.778) | 1.4884 (1.649) | −3.2246 ** (1.286) | −16.4644 *** (1.762) | −3.2295 *** (0.782) |
| City control variables | yes | yes | yes | yes | yes |
| Time fixed effect | yes | yes | yes | yes | yes |
| City fixed effect | yes | yes | yes | yes | yes |
| Firm fixed effect | yes | yes | yes | yes | yes |
| R-squared | 0.660 | 0.634 | 0.693 | 0.870 | 0.660 |
| Observations | 9133 | 9133 | 9133 | 9133 | 9133 |

Note: ***, ** and * mean significant at the level of 1%, 5% and 10%, respectively, and the values in brackets represent the robustness standard error of urban level clustering.

(b) Distinguish between industry and technology differences

This paper further draws lessons from the methods of Li and Yu [17], divides the sample of listed enterprises into high-tech enterprises and low-tech enterprises, and examines the heterogeneous impact of commercial system reform on the ESG performance behavior of high-tech and low-tech enterprises. The estimated results of its heterogeneity are shown in Table 10. Equations (1) and (2) in Table 10 are the estimated results of high-tech industry, and the estimated coefficient of commercial system reform (Policy) is significantly positive, indicating that commercial system reform has promoted the ESG performance of high-tech industry enterprises. Equations (3)–(4) are the estimated results of low-tech industries, and the estimated coefficient of commercial system reform (Policy) is positive, but it is not significant, indicating that the positive impact of commercial system reform on the ESG of low-tech enterprises is not significant. For high-tech enterprises, their R&D investment is relatively large and may involve legal issues such as intellectual property rights. The reform of the commercial system can reduce the transaction costs of high-tech enterprises in intellectual property protection and contract execution, improve market competitiveness, and help enterprises better manage internal governance and social responsibility, promoting sustainable development and the implementation of environmental protection measures. However, for enterprises in low-tech industries, the products or services of enterprises in low-tech industries are often cheap, the market competition is fierce, and the profit margin of enterprises is small, which makes the investment in the implementation of ESG relatively limited. Therefore, the positive role of commercial system reform in promoting the ESG performance of enterprises in low-tech industries has not yet emerged.

**Table 9.** Analysis results of distinguishing the nature of property rights.

| Variables | (1) State-Owned ESG | (2) State-Owned ESG | (3) Privately Operated ESG | (4) Privately Operated ESG |
|---|---|---|---|---|
| Policy | 0.0902 *** | 0.3058 *** | 0.0385 | 0.1053 |
| | (0.025) | (0.037) | (0.031) | (0.091) |
| Size | | 0.3289 *** | | 0.2563 *** |
| | | (0.039) | | (0.035) |
| Lev | | −1.1979 *** | | −0.6813 *** |
| | | (0.151) | | (0.144) |
| Cashflow | | −0.3020 | | −0.6877 *** |
| | | (0.199) | | (0.228) |
| Roa | | −0.2456 | | 0.8866 *** |
| | | (0.347) | | (0.272) |
| Board | | 0.2775 ** | | −0.0966 |
| | | (0.119) | | (0.150) |
| Indep | | 1.9390 *** | | 1.3446 *** |
| | | (0.330) | | (0.463) |
| Listage | | 0.4184 *** | | 0.0549 |
| | | (0.081) | | (0.063) |
| Constant | 4.6059 *** | −6.4335 *** | 4.2885 *** | 0.8327 |
| | (0.011) | (1.031) | (0.011) | (1.275) |
| City control variables | no | yes | no | yes |
| Time fixed effect | yes | yes | yes | yes |
| City fixed effect | yes | yes | yes | yes |
| Firm fixed effect | yes | yes | yes | yes |
| R-squared | 0.626 | 0.661 | 0.656 | 0.668 |
| Observations | 4561 | 4561 | 4572 | 4572 |

Note: *** and ** mean significant at the level of 1% and 5%, respectively, and the values in brackets represent the robustness standard error of urban level clustering.

(c) Differentiated financing constraints

Capital is an important basis for enterprises to carry out ESG practices. Financing is one of the main channels for enterprises to obtain funds, but many problems such as high financing costs are becoming more and more obvious, which has formed a greater obstacle to the ESG practice of enterprises. Therefore, the implementation of commercial system reform has had an impact on the ESG practice of enterprises with different financing constraints, which needs to be further verified. This paper uses the evaluation method of the financing constraint index to evaluate the financing ability of listed enterprises (This paper uses the size and age of enterprises that do not change much with time to construct the financing constraint index, and the specific calculation formula is SA = −0.737 × Enterprise size + 0.043 × Square of Enterprise size + 0.04 × the age of enterprises, the greater the absolute value of financing constraints, the weaker the financing constraints.). According to the measured financing constraint index value, the sample is divided into two groups, namely, lower and higher financing constraints, and the estimated results of its heterogeneity are shown in Table 11. Equations (1) and (2) in Table 11 are the estimated results of enterprises with high financing constraints, and Equations (3) and (4) are the estimated results of enterprises with low financing constraints. It is not difficult to find that the estimated coefficient of commercial system reform (policy) is significantly positive, indicating that commercial system reform has promoted the ESG performance of high-financing constraint enterprises and low-financing-constraint enterprises. However, it is worth noting that the implementation of commercial system reform has a greater promoting effect on the ESG performance of high-financing-constraint enterprises than low-financing-constraint enterprises. Compared with low-financing-constraint enterprises, high-financing-constraint enterprises have a higher demand for funds, which is relatively affected by the difficulty and high cost of financing. The reform of the commercial system can reduce the financing cost of high-financing-constraint enterprises and obtain more

marginal returns, thus better promoting the ESG performance of high-financing-constraint enterprises.

**Table 10.** Analysis results of technical differences in different industries.

| Variables | (1) Hightec ESG | (2) Hightec ESG | (3) Lowtec ESG | (4) Lowtec ESG |
|---|---|---|---|---|
| Policy | 0.0977 ** | 0.1282 ** | 0.0146 | 0.2155 |
| | (0.038) | (0.050) | (0.024) | (0.235) |
| Size | | 0.2617 *** | | 0.2294 *** |
| | | (0.041) | | (0.034) |
| Lev | | −0.9633 *** | | −0.9210 *** |
| | | (0.183) | | (0.136) |
| Cashflow | | -0.4733 | | −0.6472 *** |
| | | (0.295) | | (0.184) |
| Roa | | 0.9788 *** | | 0.0658 |
| | | (0.359) | | (0.293) |
| Board | | 0.2294 | | 0.0754 |
| | | (0.182) | | (0.114) |
| Indep | | 2.0332 *** | | 1.9487 *** |
| | | (0.596) | | (0.317) |
| Listage | | 0.0852 | | 0.0523 |
| | | (0.083) | | (0.067) |
| Constant | 4.5365 *** | −3.8473 ** | 4.4256 *** | −4.0332 *** |
| | (0.013) | (1.541) | (0.010) | (0.956) |
| City control variables | no | yes | no | yes |
| Time fixed effect | yes | yes | yes | yes |
| City fixed effect | yes | yes | yes | yes |
| Firm fixed effect | yes | yes | yes | yes |
| R-squared | 0.636 | 0.656 | 0.673 | 0.688 |
| Observations | 3322 | 3322 | 5811 | 5811 |

Note: *** and ** mean significant at the level of 1% and 5%, respectively, and the values in brackets represent the robustness standard error of urban level clustering.

(d) Distinguish industry pollution differences

According to the *Guidelines for Industry Classification of listed companies* revised by the China Securities Regulatory Commission in 2012, this paper further divides the samples of listed enterprises into enterprises in high-pollution industries and enterprises in low-pollution industries and examines the heterogeneous impact of commercial system reform on ESG performance of enterprises in high and low pollution industries. The estimated results are shown in Table 12. Equations (1) and (2) in Table 12 are the estimated results of high-pollution industries, and the estimated coefficient of commercial system reform (Policy) is significantly positive, indicating that commercial system reform has promoted the ESG performance of enterprises in high-pollution industries. Equations (3) and (4) are the estimated results of low-pollution industries, and the estimated coefficient of commercial system reform (Policy) is positive, but it is not significant, indicating that the positive impact of commercial system reform on ESG of enterprises in low-pollution industries is not significant. On the one hand, compared to high-polluting industry enterprises, low-polluting industry enterprises face relatively less environmental pressure and social responsibility requirements, and require relatively less resources to be invested in environmental protection and social responsibility. Therefore, the promotion effect of commercial system reform on ESG in low-pollution industries is relatively small. On the other hand, compared to high-polluting industry enterprises, low-polluting industry enterprises face lower risks of danger or possible government punishment (including economy, environment, policies, etc.), and have relatively lower demands for management and constraints on enterprises. Therefore, the promotion effect of commercial system reform on ESG practices of low-polluting enterprises is relatively small.

**Table 11.** Analysis results of differentiated financing constraints.

| Variables | (1) High SA ESG | (2) High SA ESG | (3) Low SA ESG | (4) Low SA ESG |
|---|---|---|---|---|
| Policy | 0.0574 ** (0.028) | 0.2578 *** (0.037) | 0.0505 * (0.029) | 0.2477 *** (0.042) |
| Size | | 0.3496 *** (0.040) | | 0.2028 *** (0.041) |
| Lev | | −0.8756 *** (0.134) | | −1.2244 *** (0.184) |
| Cashflow | | −0.6690 *** (0.187) | | −0.4687 * (0.262) |
| Roa | | 0.2889 (0.278) | | 1.0328 *** (0.366) |
| Board | | 0.2698 ** (0.129) | | 0.0333 (0.136) |
| Indep | | 2.2971 *** (0.359) | | 1.3078 *** (0.404) |
| Listage | | 0.0884 (0.088) | | 0.0856 (0.066) |
| Constant | 4.3666 *** (0.010) | −3.8245 *** (1.079) | 4.5475 *** (0.012) | −6.1677 *** (1.358) |
| Urban control variables | no | yes | no | yes |
| Time fixed effect | yes | yes | yes | yes |
| Urban fixed effect | yes | yes | yes | yes |
| Firm fixed effect | yes | yes | yes | yes |
| R-squared | 0.656 | 0.674 | 0.654 | 0.677 |
| Observations | 4567 | 4567 | 4566 | 4566 |

Note: ***, ** and * mean significant at the level of 1%, 5% and 10%, respectively, and the values in brackets represent the robustness standard error of urban level clustering.

**Table 12.** Analysis results of differentiated financing constraints.

| Variables | (1) High Pollution ESG | (2) High Pollution ESG | (3) Low Pollution ESG | (4) Low Pollution ESG |
|---|---|---|---|---|
| Policy | 0.0288 ** (0.012) | 0.2226 *** (0.047) | 0.0301 (0.024) | 0.1724 (0.134) |
| Size | | 0.2885 *** (0.045) | | 0.2429 *** (0.030) |
| Lev | | −1.0557 *** (0.166) | | −0.8855 *** (0.131) |
| Cashflow | | −0.2621 (0.282) | | −0.7149 *** (0.182) |
| Roa | | 0.1064 (0.375) | | 0.8821 *** (0.263) |
| Board | | −0.0030 (0.165) | | 0.2100 * (0.108) |
| Indep | | 1.9438 *** (0.450) | | 1.8443 *** (0.327) |
| Listage | | 0.1029 (0.097) | | 0.1245 ** (0.055) |
| Constant | 4.3864 *** (0.014) | −5.0462 *** (1.369) | 4.4755 *** (0.010) | −2.7527 *** (0.953) |
| City control variables | no | yes | no | yes |
| Time fixed effect | yes | yes | yes | yes |
| City fixed effect | yes | yes | yes | yes |
| Firm fixed effect | yes | yes | yes | yes |
| R-squared | 0.605 | 0.627 | 0.660 | 0.677 |
| Observations | 3153 | 3153 | 5980 | 5980 |

Note: ***, ** and * mean significant at the level of 1%, 5% and 10%, respectively, and the values in brackets represent the robustness standard error of urban level clustering.

## 5. Conclusions and Policy Implications

There are few articles discussing the relationship between commercial system reform and ESG performance, as well as the mediating role of green innovation in the entire ESG process. This paper focuses on the impact of commercial system reform on ESG performance. This paper regards the commercial system reform implemented in recent years as a "quasi-natural experiment". Based on the manually collected data of China's commercial system reform and the enterprise data of A-share listed companies in 2011–2021, this paper systematically examines the impact of commercial system reform on enterprise ESG performance and the intermediary role of enterprise green innovation in this process. The results show that the implementation of the commercial system reform optimizes the ESG performance of enterprises, and the green innovation of enterprises plays a positive role in the influence process. And this conclusion has passed a series of robustness tests. The ESG practices of state-owned enterprises, high-tech enterprises, high-financing-constraint enterprises, and high-pollution industry enterprises are more actively promoted by commercial system reform.

The policy implications of this paper are as follows.

Firstly, at this stage, China needs to strengthen its confidence and determination, continue to deepen the reform of the commercial system, promote the transformation of government functions, further reduce government intervention in the market, and maximize the release of institutional vitality. Especially in terms of understanding and practical ability of green development, cultivate new competitive advantages, and accelerate the construction of national sustainable development.

Secondly, we need to strengthen the complementarity of reforms such as the government power list system, better leverage government functions, prevent problems such as "shifting blame" among various departments, implement the system of handling transactions within the prescribed time in accordance with the law, effectively shorten the time required for enterprise approval, improve government efficiency, reduce the pressure of institutional transaction costs on enterprises, and optimize the business environment for enterprises.

Thirdly, enterprises should attach importance to the concept of ESG and incorporate it into their strategic planning and operational management. Enterprises should regularly disclose ESG reports and promote them to internal employees and external stakeholders through various channels to enhance their sense of social responsibility.

Fourthly, we should deepen government enterprise cooperation, establish a green cooperation mechanism between the government, enterprises, and society, establish industrial green production alliances, and jointly promote green innovation and sustainable development. We should continue to expand the green industry chain, promote the development of green finance, encourage financial institutions to invest in green industries, explore the establishment of a green credit evaluation system, and improve the financing capacity of green enterprises.

## 6. Limitations

This paper has two limitations: firstly, this paper uses text to explain the relationship between commercial system reform and enterprise ESG and has not constructed a theoretical model to explore the relationship between the two from a theoretical level. Therefore, this study can provide a reference for future related research. Secondly, there are many factors that affect the ESG of enterprises. This paper only controls a portion of them and cannot comprehensively consider the impact of other factors on the results. This also provides a reference for future related research.

**Author Contributions:** H.L.: Conceptualization, data analysis, interpretation of results, original draft. D.Y.: Investigation, writing review. Z.K.: Sample analysis, editing. All authors have read and agreed to the published version of the manuscript.

**Funding:** This study was supported by the National Social Science Fund of China (20BJL053), the Social Science Youth Fund of Ministry of Education (22YJC790053), the Central University Basic Research Business Free Project (2722023BQ054) and the Key Project of Higher Education Research of Wuhan Insitute of Technology (2022ZD02).

**Institutional Review Board Statement:** Not applicable.

**Informed Consent Statement:** Not applicable.

**Data Availability Statement:** The data that support the findings of this study are available from the corresponding author upon reasonable request.

**Conflicts of Interest:** The authors declare no competing interest.

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
