# Peer review of "Commercial System Reform, Enterprise Green Innovation and Enterprise ESG Performance"

_sustainability, doi:10.3390/su151914469_

Round 1
Reviewer 1 Report
Dear Authors,
The paper titled “Commercial system reform, enterprise green innovation and enterprise ESG performance” discusses the differential impact of policy change connected to ESG reforms in China. I have read the paper and hold the following observations:
The introduction and literature part are of merit. However, there are studies which discussed the ESG performances of state owned and non-state-owned companies in China. I recommend to the take a look at the new studies and include it in the introduction to refine your gap and in literature to enrich the literature review.
https://doi.org/10.1007/s10668-023-03636-9
https://doi.org/10.1007/s11356-023-25345-6
https://doi.org/10.1108/EMJB-09-2020-0101
Regarding methodology, the reason of choosing the methods is missing. The authors need to provide reasoning and argument on how the selected methodology is chosen. Secondly, further explanation is required for the methods used, which may include indication of strengths and weaknesses of selected model with relevant citation. The following studies may be referred for explanation of selected models.
https://doi.org/10.1007/s10668-023-03298-7
https://doi.org/10.1007/s11356-023-25345-6
In my opinion, the paper still requires a few improvements on enrichment of literature. Furthermore, methodology section needs to be focused on regarding reasoning and description.
Good luck
Dear authors,
English language seems okay, however, it required a minor grammer check.
Reviewer 2 Report
The aim of this manuscript to investigate the relationship between commercial system reform and corporate ESG performance, and analyzes the intermediary effect of corporate green innovation between them, basse on China's listed companies from 2011-2021.
Our recommendation to improve this manuscripts as follows:
1. Add new paragraph in the introduction section to structure the manuscript sections. ( e.i: this paper is structred as follows :.............section 2, section3.......etc).
2. Theoretical analysis and research assumptions:
* Structure: Each paragraph is compsed max ( 5-6 lines): try to prepare this Fomr to all the paper sections.
* Cited references : change these forms of citation by using Brackets [..] ( Chen & Wang, 2015; Yang & Gao, 2014; Li, 2021, .... etc.). This paper in Sustainability can help you to edit again your cited references. https://doi.org/10.3390/su132212438 (you can add in yur literature review if you see important)
* Literature review: you must update from 2021-2023( new recent cited references from Sustainability, Economie, Administrave science Databases MDPI, ScienceDirect, Springer Nature, Elsevier, etc.). This can give your study more actuel theorical significance. You can add comparaision theory Studies in table, as option. We can help your theory farmework with these related research papers (https://doi.org/10.3390/su14095160 ; https://doi.org/10.1016/j.jclepro.2021.128830 ; https://doi.org/10.3390/su132212438 ; https://doi.org/10.3390/su13179819 )
3. Model building, variables and data:
* Add cited reference Authors who used the Conceptuel MODEL ( Model building, This paper uses a multi period dual difference model for empirical evaluation, and the specific model is constructed as follows)
* Variable selection : It will more presented, if you join defintion variables tables ( Variables, period of studies, Reference authors, Aime of study).
* Table 1. descriptive statistics of main variables. : for all tables add resource and statistaical software used).
4.Empirical analysis :
* Robustness test : Provide the theorical significance of this Test, than compare with your findings.
* A placebo test: ( the same)
* ''With the introduction of control variables, after controlling other factors, the direction and significance of the reform of commercial system have not changed,'' : Could you explain more?
* Table 4. PSM-DID region results.: Add the P-value significance note at the end of the tables. ( for each table do not miss this statistical significance note)
* Testing the intermediary mechanism of enterprise green innovation :
This paper uses the number of green
* invention patent registrations of enterprises and R&D investment of enterprises to measure: Why you just used, there're other green innovation variables to use?
* Table 5. impact mechanism test of green innovation capability of enterprises: What is your analysis about this results provided in this table?
* Heterogeneity analysis : Provide the theorical significance of this Test, than compare with your findings.
* Table 6. analysis results of distinguishing the nature of property rights: Missed analysis?
* Table 7- table 9 : all these tables missed statistical analysis.
6. Discussion section : It is very important section to discuss your results/ findings?
7. Limitations: is missed
8. References:
*DOI is messed for all cited references.
* Use the reference model of SUSTAINABILITY Journal (see: this paper to help you to edit your references section, if it is usefull join it in your literature review: https://doi.org/10.3390/su14106030)
--
G Luck.
Grammar and spelling to check.
Reviewer 3 Report
1. Why you picked up the time period from 2011 to 2021? As we all know the Covid-19 pandemic, you must conduct a robustness check by removing all the data after the year 2020.
2. You must add one section to clearly provide the institutional background of commercial system reform. In particular, you must explain why this reform is important with sufficient evidence.
3. You must rewrite the introduction since it is too long and contains so many “light” statements, which means you have no evidence or support for the statements. For example, “At present, China is in a new stage of economic development.” What does that mean? Any academic support? It is more like some political statement instead of meaningful results. The same light statement can be found like “Enterprises not only need to improve their own development quality and management efficiency, but also need to pay more attention to environmental protection, reduce carbon emissions and focus on longterm sustainable development.” Again, what is the support or evidence?
I can list more examples found in the introduction. Please rewrite the introduction to make sure every sentence is academically meaningful.
4. You should clearly motivate your question within the first two paragraphs, while you must shorten your first two paragraphs. Actually, most of the paragraphs in the paper are too long to be readable.
5. You must rewrite the literature and hypothesis part. You must first read sufficiently number of papers published in international peer-reviewed journals and then write your literature and hypothesis part. Your current version is mainly based on numerous papers written in Chinese. That’s not acceptable for an international journal. If there is no relevant papers in decently good international journals, the reasonable inference is that your research question is not academically important.
In addition, you must organize the theoretical argument in a better way. Currently, your argument is not convincing since the logic is not clear enough.
6. Which ESG index was used in the paper? In the paper, you said, “This paper uses the China Securities ESG index to quantitatively evaluate the ESG performance of enterprises in benchmark regression, robust regression, intermediary mechanism test and heterogeneity analysis. There are three reasons why this paper chooses Huazheng ESG score as the performance of enterprise ESG performance”.
Are you using China Securities ESG index or Huazheng ESG score? It is too confusing.
In addition, I’m not convinced by your argument on choosing Huazheng ESG score. All the three reasons you listed are light statement. You must either cite some papers published in decently good international journals to support your argument or provide detailed explanations to show the advantage of Huazheng ESG.
Also, since there are two ESG data sources you can use, you must provide a robustness check using another ESG data sources.
7. “This article refers to Huang et al. (2020)”. This is related to the essentials of your empirical work design, then you must cite an important paper published in a decently good international journal. The paper you cited is not even in a top journal in China. In addition, the journal name is Journal of Finance and Economics instead of “Financial Research”. Here is the link https://qks.shufe.edu.cn/J/CJYJ/Article/Details/6ba3cc5e-601b-471a-99a1-5a72023b4f09
8. I have never heard a model named “multi period dual difference model”. Please check it carefully and correct the name.
9. You must provide the details of how you processed the data instead of citing a paper published in a Chinese journal. Also, what is “a” in “Wang & Lu (2019 a)”? In the detailed process, you must clearly state the steps of data processing and the resulted number of observations after each step.
10. The policy implications should be better related to the findings.
11. You should add one subsection to explain the limitations of the current work.
This is a paper translated from some Chinese paper. The authors must further polish the language with the help from some language editing service. The current version is hard to follow.
Round 2
Reviewer 2 Report
1. Literature review: you must update from 2021-2023( new recent cited references from Sustainability, Economie, Administrave science Databases MDPI, ScienceDirect, Springer Nature, Elsevier, etc.). This can give your study more actuel theorical significance. You can add comparaision theory Studies in table, as option. Add these related papers to your theory farmework and in your references section (https://doi.org/10.3390/su14095160 ; https://doi.org/10.1016/j.jclepro.2021.128830 ;
2. * Heterogeneity analysis : Provide the theorical significance of this Test, than compare with your findings.
* Table 6. analysis results of distinguishing the nature of property rights: Missed analysis?
3. Validation of the theoritical hypotheses? At theendof your discussion section, present a significant table of your hypotheses (H1, H2). It means supported or no.
Good luck.
Minor editing english language required.
Reviewer 3 Report
The authors have addressed my concerns well.
Author Response
Thank you very much for the valuable opinions of the reviewers. Please sign on the review report. Thank you.
Round 3
Reviewer 2 Report
1. citations: all cited references must be in brackets [..]
To explain more for examples: ( remore the cited references, and keep just the number between bracktes). See the Author's Guidline of the journal sustainability.
(Baker et al., 2021) [1].
... etc.
(Zahid et al., 2023) [6] ,
(Cai et al., 2016) [7] ; etc.
2. Please revise again:
Full text concerning the errors and spellings.
Good luck.
Please revise again:
Full text concerning the errors and spellings.
